# ADMIRE++: Explainable Anomaly Detection in the Human Brain via Inductive Learning on Temporal Multiplex Networks

**Ali Behrouz** [1]  **Margo Seltzer** [1]

## Abstract

Understanding the human brain is an intriguing goal for neuroscience research. Due to recent advances in machine learning on graphs, representing the connections of the human brain as a network has become one of the most pervasive analytical paradigms. However, most existing graph machine learning-based methods suffer from a subset of three critical limitations: They are ① designed for one type of data (e.g., fMRI or sMRI) and one individual subject, limiting their ability to use complementary information provided by different images, ② designed in supervised or transductive settings, limiting their generalizability to unseen patterns, ③ blackbox models, designed for classifying brain networks, limiting their ability to reveal underlying patterns that might cause the symptoms of a disease or disorder. To address these limitations, we present ADMIRE, an inductive and unsupervised anomaly detection method for multimodal brain networks that can detect anomalous patterns in the brains of people living with a disease or disorder. It uses two different casual multiplex walks, *inter-view* and *intra-view*, to automatically extract and learn temporal network motifs. It then uses an anonymization strategy to hide node and relation type identities, keeping the model inductive. We then propose a simple, tree-based explainable model, ADMIRE++, to explain ADMIRE predictions. Our experiments on Parkinson's Disease, Attention Deficit Hyperactivity Disorder, and Autism Spectrum Disorder show the efficiency and effectiveness of our approaches in detecting anomalous brain activity.

[1]Department of Computer Science, University of British Columbia, Vancouver, BC, Canada. Correspondence to: Ali Behrouz <alibez@cs.ubc.ca>.

*Workshop on Interpretable ML in Healthcare at International Conference on Machine Learning (ICML)*, Honolulu, Hawaii, USA. 2023. Copyright 2023 by the author(s).

## 1. Introduction

Recently, the fields of neuroscience and brain imaging research have undergone a significant shift in focus from region-specific analyses to network models (Bassett & Sporns, 2017; Mišić & Sporns, 2016), largely due to the rapid development of modern neuroimaging technology. Network models of the brain represent regions of interest (ROIs) as nodes and calculate pairwise similarities between regions to form edges (Finn et al., 2015), usually derived from functional Magnetic Resonance Imaging (fMRI) or structural Magnetic Resonance Imaging (sMRI). These models have demonstrated their effectiveness in enhancing our understanding of brain diseases and disorders (Chatterjee et al., 2021; Preti et al., 2017). As a result, empirical data on brain networks has substantially increased in size and complexity, leading to a strong demand for appropriate tools and methods to model and analyze this data (Preti et al., 2017).

At the same time, there has been significant interest in machine learning methods for analyzing graph-structured data in various domains, such as drug discovery (Xiong et al., 2019), neuroscience (Abrate & Bonchi, 2021), and biology (Gao et al., 2023). While several studies demonstrated the effectiveness of machine learning on graphs for analyzing human brain networks, most focus on graph or node classification tasks (Kan et al., 2022c; Cui et al., 2022b). These tasks involve detecting diseases (Zhu et al., 2022a), predicting biological features (Kan et al., 2022c), and identifying functional systems (Behrouz & Hashemi, 2022). However, detecting abnormal brain activity in people with neurological disorders is a crucial step in understanding the causal mechanisms of symptoms, facilitating early detection, and developing medical treatments. Most existing studies consider a single brain network (from a single type of neuroimage or a single subject), which can be noisy or incomplete (Agrawal et al., 2020; Zhang et al., 2020). To address this limitation, De Domenico (2017) suggests using static multiplex networks. Multiplex networks are graphs where nodes can be connected by different types of edges (Kivelä et al., 2014; Hashemi et al., 2022). Edge types can be the brain network of different subjects (Behrouz et al., 2022a) or different neuroimaging modalities (Zhu et al., 2022b) (see §3.1).

**Limitation of Previous Methods.** Although anomaly detection in graphs is a well-studied problem, brain networks have five unique traits that make directly applying existing graph anomaly detection models impractical: ① Noisy data: a single neuroimaging data sample can be extremely noisy and inaccurate (Agrawal et al., 2020), which hinders the identification of biological insights into the structure of brain networks. Existing general anomaly detection methods can use only a single brain image, making them sensitive to noise, or one must aggregate different neuroimages as a pre-processing step, missing complex brain activity in each brain image. ② Multimodal neuroimaging: while several studies discussed the importance of using different neuroimage types (e.g., fMRI, sMRI, etc.), because different modalities provide complementary information (Zhang et al., 2018c; Zhu et al., 2022b), existing works are limited to a single type of neuroimages and are unable to incorporate information about different modalities. ③ Complex activity: brain activities are complex and potentially different in different subjects, while existing methods are designed in the transductive setting, which limits their generalizability to unseen nodes or patterns. ④ Time alignment: existing methods assume that the timestamps in different graphs are meaningfully related. However, while modeling neuroimage data as *temporal* brain networks, the timestamps might be shifted and are unlikely to be aligned across brain images of different subjects. ⑤ Explainability: decision making on health-related data, which is sensitive, requires explainable models, but existing methods are uninterpretable black boxes.

There are two other limitations that plague existing studies: ⓘ These studies assume pre-defined anomaly patterns or man-made features. Such approaches do not easily generalize to the brain activity of different individuals. Moreover, in a real-world scenario, brain activity might be more complex in nature, and it is nearly impossible to detect anomalies with high accuracy using pre-defined patterns/roles. ⓘⓘ These methods are designed for static brain networks, missing the dynamics of brain activity over time.

To mitigate the limitations, we introduce ADMIRE (**A**nomaly **D**etection in **M**ultiplex **Br**ain **Ne**tworks). ADMIRE uses two novel temporal walks, *inter-view* and *intra-view* walks, to capture the causal relationships between brain activities across different views and within a single view, respectively, over time. Next, it uses an anonymization method based on the correlation between network motifs to hide the identity of nodes and views, keeping the model inductive during training. To overcome noise in the data and/or to take advantage of complementary information provided by different neuroimage modalities, ADMIRE uses a new attention mechanism to incorporate the node encodings obtained from different views. To mitigate the time alignment issue, we use a non-periodic time encoding module

that encodes each timestamp. To learn the structural and temporal properties of the network, ADMIRE encodes the information about each walk by mixing the encoding of the sequence of nodes that appears in the walk, along with their timestamps, via an MLP-Mixer (Tolstikhin et al., 2021). Finally, we design a post-hoc decision-tree-based explanation method to explain ADMIRE's predictions based on its extracted motifs. Experimental evaluation shows the superior performance of ADMIRE over baselines, and the importance of ADMIRE's critical components. We use real-world datasets to show how ADMIRE can be used to detect abnormal brain activities in a control group with brain disease or disorder.

## 2. Related Work

**Anomaly Detection in Brain Networks.** Several recent studies focus on analyzing brain networks to distinguish healthy and diseased human brains (Jie et al., 2016; Chen et al., 2011; Wee et al., 2011). Due to the success of GNNs in analyzing graph-structured data, deep models have been proposed to predict brain diseases by learning the brain network structure (Kan et al., 2021; Cui et al., 2021; Kan et al., 2022a; Zhu et al., 2022a; Cui et al., 2022b). All these methods are designed for graph or node classification and cannot directly be extended to edge-anomaly detection. Also, several anomaly detection methods have been proposed to find anomalous regions or subgraphs in the brain, which might indicate the presence of a disease (Chatterjee et al., 2021; Zhang et al., 2016; Liu et al., 2020). All these methods are designed for node or subgraph anomaly detection tasks in *single* brain networks and cannot easily be extended to detect anomalous edges in *multiplex* brain networks. Also, these methods are not learning-based and consider only pre-defined patterns/rules for anomalies.

**Anomaly Detection in Multiplex Networks.** Several non-machine learning methods for anomaly detection in static multiplex networks have been proposed based on eigenvector centrality (Mittal & Bhatia, 2018), clique/near-clique structures (Bindu et al., 2017), multi-normality (Bansal & Sharma, 2020), node centrality (Maulana & Atzmueller, 2020), and persistence summary (Ofori-Boateng et al., 2021). These models are not able to learn from data and are limited to pre-defined rules/patterns. To address this issue, recently, learning-based methods have been proposed. ANOMMAN (Chen et al., 2022) uses an auto-encoder module and a GCN-based decoder to detect node anomalies in static multiplex networks. All of these approaches are limited to static multiplex networks and are designed to detect topological anomalous subgraphs, nodes, or events, and cannot identify anomalous edges. The only exception is ANOMULY (Behrouz & Seltzer, 2022), a GNN-based anomaly detection method in multiplex networks. However,

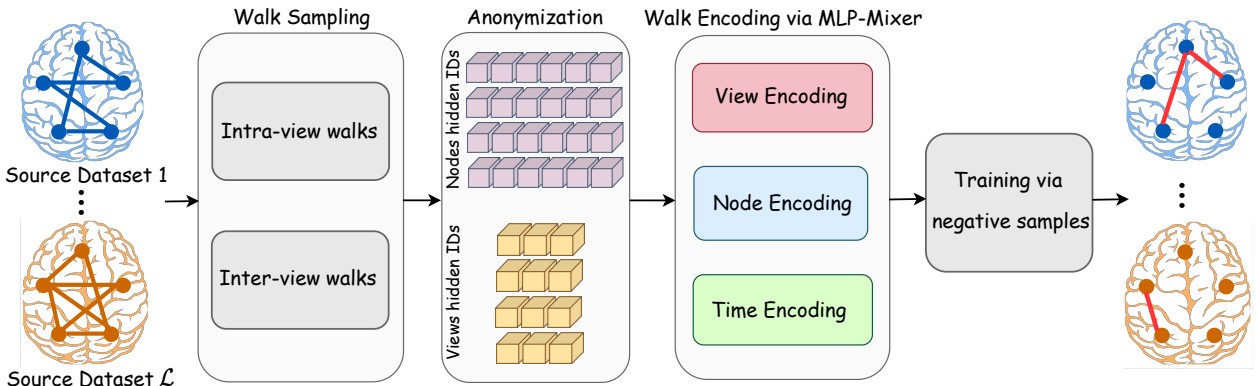

*Figure 1.* **Schematic of the ADMIRE model**. ADMIRE consists of four main stages called (1) Walk Sampling, (2) Anonymization, (3) Walk Encoding, and (4) Training via generating negative samples.

it is designed for the transductive setting and cannot scale to multiplex networks with a large number of views (see §4.1).

Additional related work is in Appendix B.

## 3. Methods

**Definition 3.1** (Temporal Multiplex Networks). A temporal multiplex network $\mathcal{G} = \{G_r\}_{r=1}^{\mathcal{L}} = (\mathcal{V}, \mathcal{E}, \mathcal{X})$, can be represented as a sequence of connections with different types that arrive over time, i.e., $\mathcal{E} = \{(e_1, t_1), (e_2, t_2), \dots\}$, where $\mathcal{V}$ is the set of nodes, $\mathcal{L}$ is the set of relation types, $\{e_1, e_2, \dots\} \subseteq \mathcal{V} \times \mathcal{V} \times \mathcal{L}$, and $\mathcal{X} \in \mathbb{R}^{|\mathcal{V}| \times f}$ is a matrix that encodes node attribute information for nodes in $\mathcal{V}$. Given a relation type $r$, we use $G_r = (\mathcal{V}, \mathcal{E}_r, \mathcal{X})$ to denote the corresponding graph of the relation type $r$ (a.k.a the $r$-th view of the graph), and we denote the set of vertices in the neighborhood of $u \in \mathcal{V}$ in relation $r$ as $\mathcal{N}_r(u)$. Given time $t$, we use $\mathcal{E}_r^t(u) = \{(e, t') \in \mathcal{E}_r | u \in e \text{ and } t' \leq t\}$ to represent the set of connections attached to a node $u$ in relation type $r$ before a given time $t$.

Our goal is to detect anomalous incoming edges. Specifically, given the current time $t_{\text{now}}$, for each edge $e = (u, v, r, t_{\text{now}}) \in \mathcal{E}$, we produce an anomaly score $\varphi(e)$.

### 3.1. Modeling Neuroimages as Multiplex Brain Network

To take advantage of complementary information provided by different modalities of neuroimaging, neuroimages of different subjects with the same disease/disorder, and different frequency band filters, we focus on three ways to model neuroimaging data as multiplex networks:

① **Activity in different frequency bands**: previous works on fMRI images utilize filtering procedures to extract signals within a particular frequency range, typically between 0.01

and 0.1 Hz (De Vico Fallani et al., 2014). However, the selection of the frequency band carries significant implications for the functional representation of the brain. De Domenico et al. (2016) shows that brain signals in a range between 0.01 and 0.25 Hz, in steps of 0.02 Hz provide unique information and should be neither aggregated nor neglected. We suggest using multiplex brain networks, where each view represents the correlations graph of signals in a specific range.

② **Multimodal brain networks**: Several studies discussed the importance of using different neuroimage types (e.g., fMRI, sMRI, Diffusion Tensor Imaging (DTI), etc.) in brain network analysis as different modalities of brain networks provide complementary information (Zhang et al., 2018c; Zhu et al., 2022b). In this case, a multiplex brain network is a multimodal brain network, where each view represents the obtained brain network from a specific type of neuroimage.

③ **Different subjects**: To mitigate the existence of noise in a brain network generated from an individual (Lanciano et al., 2020; Zhang et al., 2020), most existing methods aggregate (e.g., averaging) the data from different individuals (Chatterjee et al., 2021). However, this aggregation discards complex patterns in each individual's brain activities. Moreover, it is known that individuals having the same disease or disorder share similar patterns (Kan et al., 2022b), which means that disorder/disease-specific anomalous activities require consideration of the brain networks of different subjects. We suggest using multiplex brain networks, where each view represents the brain network of an individual.

### 3.2. Anonymous Multiplex Temporal Walk

When modeling neuroimage data as multiplex networks, there are two advantages: ① complementary information from different views enhances the effectiveness and robustness of learning brain activities, and ② capturing causal ef-

fects between different views improves performance. However, a major challenge in designing machine learning models for multiplex brain networks is learning the type of data modeling. We address this challenge by automatically learning causal effects between activities in different neuroimages and incorporating complementary information. We propose two temporal multiplex walks to capture causal effects and dynamics in different views over time. These walks are combined using a learnable neural layer to automatically determine their importance. Additionally, we define an anonymization process to preserve model inductiveness during training. Our approach leverages multiplex temporal walks as proxies for temporal motifs in multiplex networks, extracting the causality of edge existence within a specific view and across different views.

**Inter-view Temporal Walk.** To capture the correlation between different views and extract the causality of an edge from different views of the network, in the inter-view temporal walk, we let the walker walk across views. Accordingly, an inter-view temporal walk $W_{\text{inter}}$ on temporal multiplex networks can be represented as:

$$W_{\text{inter}} = ((u_0, r_0, t_0), (u_1, r_1, t_1), \ldots, (u_m, r_m, t_m)),$$

where $t_0 \geq t_1 \geq \cdots \geq t_m$ and $(u_i, u_{i+1}, r_{i+1}, t_{i+1}) \in \mathcal{E}$. That is, the walker walks through time capturing temporal causality, and it walks over different views to capture dependencies of connections in different views; this latter ability leverages the complementary information provided by different relation types (e.g., fMRI and sMRI). We let $W_{\text{inter}}(i)$ denote the $i$-th element of the temporal multiplex walk, $(u_i, r_i, t_i)$. Also, we use $W_{\text{inter}}(i,0), W_{\text{inter}}(i,1)$, and $W_{\text{inter}}(i,2)$ to refer to $u_i, r_i$, and $t_i$, respectively.

**Intra-view Temporal Walk.** When there is no causal relationship between different types of interactions (e.g., brain networks of different subjects), we limit our walks to a specific type of connection. Given a type of relation $r$, an intra-view temporal walk $W^r_{\text{intra}}$ on a view of a temporal multiplex network can be represented as:

$$W^r_{\text{intra}} = ((u_0, r, t_0), (u_1, r, t_1), \ldots, (u_m, r, t_m)),$$

where $t_0 \geq t_1 \geq \cdots \geq t_m$ and $(u_i, u_{i+1}, r, t_{i+1}) \in \mathcal{E}$. However, different views still provide complementary information (see §3.1). To take advantage of this complementary information, in §3.3, we design an attention mechanism that incorporates the information of different connection types.

**How to sample a temporal multiplex walk?** Newer connections are usually more informative than older connections (Wang et al., 2021; Jin et al., 2022; Behrouz et al., 2023). Therefore, we use a biased sampling method with hyperparameter $\mu$ to control the importance of recent connections. Given the time of a previously sampled edge,

$t_0$, we sample an adjacent edge at time $t$ with probability proportional to $\exp\left(\mu(t - t_0)\right)$. In multiplex networks, the correlation of different pairs of views can be different (Park et al., 2020; Behrouz & Hashemi, 2022) and for connections in a given view $r$, a subset of views might play more important roles in causality extraction. Accordingly, in inter-view temporal walks, we use a biased sampling method and sample link $(u'_1, u'_2, r', t')$ after previously sampled link $(u_1, u_2, r, t)$ with probability proportional to $\psi(r, r')$. In fact, $\psi(r, r')$ shows the importance of view $r$ for view $r'$. In §3.3, we discuss how to calculate $\psi(r, r')$. See Appendix C for the pseudocode.

Given a (potential) link $(u, v, r, t)$, we use the above procedure to generate $M$ inter-view and $M'$ intra-view walks with $m$ steps starting from each of nodes $u$ and $v$. We use $\mathcal{S}_{\text{inter}}(u)$, $\mathcal{S}_{\text{intra}}(u)$, $\mathcal{S}_{\text{inter}}(v)$, and $\mathcal{S}_{\text{intra}}(v)$ to store started walks from $u$ and $v$, respectively.

**Anonymization Process.** Micali & Zhu (2016) studied Anonymous Walks (AWs), which replace a *node's identity* by the order of its appearance in each walk. The main limitation of AWs is that the position encoding of each node depends only on its specific walk, missing the dependency and correlation of different sampled walks (Wang et al., 2021). To mitigate this drawback, Wang et al. (2021) suggest replacing node identities with the hitting counts of the nodes based on a set of sampled walks, capturing the correlation between different walks (Wang et al., 2021; Jin et al., 2022; Behrouz et al., 2023). In multiplex networks, we need to hide the identity of both nodes and views (e.g., relation types) to keep the model inductive. Given a (potential) link $(u, v, r, t)$, let $w_0 \in \{u, v\}$. To capture the correlation across different walks, which is a key to reflecting the network dynamics, for a given node $w$ that appears on at least one walk in $\mathcal{S}_{\text{inter}}(u) \cup \mathcal{S}_{\text{inter}}(v)$, we use a relative vector $\mathcal{C}(\mathcal{S}_{\text{inter}}(w_0), w) \in \mathbb{Z}^{m+1}$ that represents the number of times in $\mathcal{S}_{\text{inter}}(w_0)$ that node $w$ appears at certain positions. That is,

$$\begin{aligned}
&\mathcal{C}_i\left(\mathcal{S}_{\text{inter}}(w_0), w\right) = \\
&\left|\{W_{\text{inter}} | W_{\text{inter}} \in \mathcal{S}_{\text{inter}}(w_0), w = W_{\text{inter}}(i, 0)\}\right|,
\end{aligned}$$

for $0 \leq i \leq m$. Similarly, we define $\mathcal{C}(\mathcal{S}_{\text{intra}}(w_0), w)$ over intra-view temporal walks. Now, we assign a hidden identity to node $w$, ID($w$), as the set of $\mathcal{C}_i\left(\mathcal{S}_{\text{inter}}(w_0), w\right)$ and $\mathcal{C}_i\left(\mathcal{S}_{\text{intra}}(w_0), w\right)$.

Given a set of walks (e.g., $\mathcal{S}_{\text{inter}}(w_0)$), we count the number of times we see a relation type at certain positions when we start from a specific relation type to capture the correlation of different views. For a given relation type $r$, we use a relative vector $\mathcal{C}^{\text{view}}(\mathcal{S}_{\text{inter}}(w_0), r) \in \mathbb{Z}^{m+1}$ that counts number of times that a relation with type $r$ appears at certain

positions in $\mathcal{S}_{\text{inter}}(w_0)$:

$$\mathcal{C}_i^{\text{view}}\left(\mathcal{S}_{\text{inter}}(w_0), r\right) =$$
$$|\{W_{\text{inter}}|W_{\text{inter}} \in \mathcal{S}_{\text{inter}}(w_0), r = W_{\text{inter}}(i, 1)\}|,$$

for $0 \leq i \leq m$. Accordingly, we use $\text{ID}^{\text{view}}(r) = \{\mathcal{C}^{\text{view}}\left(\mathcal{S}_{\text{inter}}(u), r\right), \mathcal{C}^{\text{view}}\left(\mathcal{S}_{\text{inter}}(v), r\right)\}$ to hide the identity of view $r$. Note that, although intra-view walks are within a single view, we still need to hide the identity of the view and we use the same $\text{ID}^{\text{view}}(r)$ as above.

### 3.3. Neural Encoding

Most existing methods on walk encoding see a walk as a sequence of vertices and uses sequence encoders such as RNNs or TRANSFORMERs to encode each walk. The main drawback of these methods is that they fail to directly process temporal walks with irregular gaps between timestamps. That is, sequential encoders can be seen as discrete approximations of dynamic systems; however, this discretization often fails if we have irregularly observed data (Kidger et al., 2020). We present a neural network to encode temporal multiplex walks so that we can extract structural and temporal information from the network with continuous time dyanamic. The process consists of ① time encoding module to encode the time, ② node encoding module to encode the position of vertices, ③ view encoding to encode relation type, and ④ walk encoding to encode each extracted motifs.

**Time Encoding.** Existing methods in temporal graph learning (Cong et al., 2023; Wang et al., 2021) use random Fourier features (Kazemi et al., 2019) to encode time. However, this approach captures only periodicity in the data, while in brain activity patterns we also need to learn non-periodic patterns dependent on the progression of time (e.g., in task-based fMRI). To this end, we also add a learnable linear term to the feature representation of time encoding. That is, we encode a given time $t$ as:

$$\mathscr{T}(t) = (\boldsymbol{\omega}_l t + \mathbf{b}_l) \,||\, \cos(t\boldsymbol{\omega}), \tag{1}$$

where $\boldsymbol{\omega}_l, \mathbf{b}_l \in \mathbb{R}$ and $\boldsymbol{\omega} \in \mathbb{R}^d$ are learnable parameters, and $||$ denotes concatenation.

**Node Encoding.** We define a node encoding function $\zeta(.)$ that encodes each node $w$ based on $\text{ID}(w)$. However, since the concept and task of intra-view and inter-view walks are different, we first break the $\zeta(.)$ function over these walks, called $\zeta_{\text{intra}}(.)$ and $\zeta_{\text{inter}}(.)$, respectively, and then interpolate between them by a learnable parameter $\lambda$ to obtain $\zeta(.)$.

For each node $w$ that appears on at least one walk in $\mathcal{S}_{\text{inter}}(u) \cup \mathcal{S}_{\text{inter}}(v)$, we use *one* simple MLP to encode the $w$'s hidden identities:

$$\zeta_{\text{inter}}(w) = \text{MLP}\left(\mathcal{C}(\mathcal{S}_{\text{inter}}(u), w)\right) + \text{MLP}\left(\mathcal{C}(\mathcal{S}_{\text{inter}}(v), w)\right). \tag{2}$$

While inter-view walks naturally capture the causal relationship and correlation between different types of connections, intra-view walks capture causality within one type of connection. To take advantage of complementary information in multiplex networks, we need to aggregate the information provided by inter-view walks in different views. However, the importance of views might be different (e.g., one disease might be more correlated with functional connectivity than structural connectivity). We design an attention mechanism that learns the importance of each view for other views. Given two arbitrary views $r_1, r_2$, let $\eta(r_1)$ and $\eta(r_2)$ be the learned encoding of $r_1$ and $r_2$. The importance of $r_2$ for $r_1$, $\psi(r_1, r_2)$, is defined as:

$$\psi(r_1, r_2) = \frac{\exp\left(\sigma\left(\vec{a}^T.[\mathbf{W}^{\text{att}}\eta(r_1) \,||\, \mathbf{W}^{\text{att}}\eta(r_2)]\right)\right)}{\sum_{r' \in \mathcal{L}} \exp\left(\sigma\left(\vec{a}^T.[\mathbf{W}^{\text{att}}\eta(r_1) \,||\, \mathbf{W}^{\text{att}}\eta(r')]\right)\right)},$$

where $\vec{a}$ and $\mathbf{W}^{\text{att}}$ are learnable parameters and $\sigma(.)$ is an activation function (e.g., ReLU). Given a relation type $r' \in \mathcal{L}$, we define view-based node encoding $\zeta_{\text{intra}}^{r'}(w)$ as:

$$\zeta_{\text{intra}}^{r'}(w) = \text{MLP}\left(\mathcal{C}(\mathcal{S}_{\text{intra}}^{r'}(u), w)\right) + \text{MLP}\left(\mathcal{C}(\mathcal{S}_{\text{intra}}^{r'}(v), w)\right).$$

Next, we aggregate these node embeddings to incorporate information from different views and obtain $\zeta_{\text{intra}}(w)$:

$$\zeta_{\text{intra}}(w) = \sum_{r' \in \mathcal{L}} \psi(r, r')\zeta_{\text{intra}}^{r'}(w). \tag{3}$$

Now, we use a learnable parameter $\lambda$ to automatically learn the importance of each $\zeta_{\text{intra}}(w)$ and $\zeta_{\text{inter}}(w)$ based on the data. This formulation lets our model learn to interpolate between Equation 2 and Equation 3, which enables it to be flexible in each way the neuroimaging data is modeled (§3.1). Therefore, $\zeta(w)$ is defined as:

$$\zeta(w) = \zeta_{\text{intra}}(w) + \lambda \times \zeta_{\text{inter}}(w).$$

When there is no causal relation between different views (e.g., when views are brain networks of different subjects), our model is expected to set $\lambda \approx 0$ (see §4.1).

**View Encoding.** For each view $r \in \mathcal{L}$, we use *one* simple MLP to encode $r$'s hidden identities:

$$\eta(r) = \text{MLP}\left(\mathcal{C}^{\text{view}}(\mathcal{S}_{\text{inter}}(u), r)\right) + \text{MLP}\left(\mathcal{C}^{\text{view}}(\mathcal{S}_{\text{inter}}(v), r)\right).$$

**Walk Encoding.** Given a walk $\hat{W} \in \{W_{\text{inter}}, W_{\text{intra}}\}$, we use node encoding function $\zeta(.) : \mathbb{Z}^{(m+1) \times 4} \to \mathbb{R}^{k_1}$ to encode hidden node identities and $\eta(.) : \mathbb{Z}^{(m+1) \times 2} \to \mathbb{R}^{k_2}$ to encode hidden view identities. We then concatenate their outputs with the embedding of the node's corresponding timestamp. Finally, we use an MLP-Mixer (Tolstikhin et al., 2021) to mix these encodings to obtain the walk encoding:

$$\text{ENC}(\hat{W}) =$$
$$\text{MEAN}\left(\mathbf{H}_{\text{token}} + \mathbf{W}^{(2)}\sigma\left(\text{LayerNorm}(\mathbf{H}_{\text{token}})\mathbf{W}^{(1)}\right)\right),$$

where the $i$-th row of $\mathbf{H}_{\text{token}}$ is the concatenation of $\zeta\left(\text{ID}\left(\hat{W}(i,0)\right)\right), \eta\left(\text{ID}^{\text{view}}\left(\hat{W}(i,1)\right)\right)$ and $\mathscr{T}(t_i)$. In the above equations, $\mathbf{W}^{(1)}$ and $\mathbf{W}^{(2)}$ are learnable parameters, LayerNorm is layer normalization (Ba et al., 2016) and $\sigma(.)$ is a nonlinear function (e.g., Gaussian error linear units, GeLU (Hendrycks & Gimpel, 2020)).

**Anomaly Score.** Given a link $e = (u, v, r, t) \in \mathcal{E}$, we sample temporal multiplex walks and then encode each walk $W \in \mathcal{S}_{\text{inter}}(u) \cup \mathcal{S}_{\text{inter}}(v) \cup \mathcal{S}_{\text{intra}}(u) \cup \mathcal{S}_{\text{intra}}(v)$ as described above. Next, we use mean-pooling to aggregate walks' encodings and encode link $e$. Finally, we use a 2-layer perceptron to make the anomaly score:

$$\varphi(e) = \text{MLP}\left(\frac{1}{M+M'}\sum_{\hat{W}}\text{ENC}(\hat{W})\right), \qquad (4)$$

where $M$ and $M'$ are the numbers of inter-view and intra-view walks.

**Negative Sample Generator.** We generate negative samples to train ADMIRE in an unsupervised manner. Previous anomaly detection methods mostly use (simple or biased) random negative samples (Zheng et al., 2019; Behrouz & Seltzer, 2022), which limit their generalizability to real anomalous patterns (Poursafaei et al., 2022). Moreover, these methods are designed for simple networks and cannot generalize to anomalous patterns in multiplex networks (see §4.1). Inspired by Poursafaei et al. (2022), we design a novel negative sampling method for temporal *multiplex* networks.

Let $\mathcal{E}_{\text{train}}$ and $\mathcal{E}_{\text{t}}$ be the set of edges in the training set and in timestamp $t$, respectively. For each edge in the training set $e = (u, v, r, t) \in \mathcal{E}$, we generate three types of negative samples: ① Inter-view negative samples: We use these negative samples so our model learns to detect connections that are anomalous across different views. We randomly generate a negative connection with relation type $r$ with probability inversely proportional to the number of views in which this connection appears. The intuition is that if two nodes are already connected with several types of connections, a connection of yet another type is unlikely to be an anomalous connection. ② Intra-view negative samples: Here, we follow previous negative sampling generation methods (Zheng et al., 2019; Behrouz & Seltzer, 2022) and randomly change one endpoint of a connection to another node and keep the type of connection unchanged. ③ Historical negative samples: we generate negative edges from the set of edges that have been observed during previous timestamps but are absent in the current timestamp. That is, we randomly sample an edge $e \in \mathcal{E}_{\text{train}} \cap \bar{\mathcal{E}}_{\text{t}}$.

**Training and Loss Function.** Let $\mathcal{E}_{\text{train}}$ be the set of edges

in the training set and $\mathcal{E}_{\text{neg}}$ be the set of generated negative samples. For each link $e \in \mathcal{E}_{\text{train}} \cup \mathcal{E}_{\text{neg}}$ we generate temporal multiplex walks to find view-aware edge encoding of $e$. Next, we use the margin-based pairwise loss (Bordes et al., 2013) to train the model. To avoid overfitting, we also use an $L2$-regularization loss, $\mathscr{L}_r^{reg}$, which is the summation of the $L2$ norm of all trainable parameters.

### 3.4. ADMIRE++: Post-hoc Explanation of ADMIRE

Prediction and decision-making on brain networks is a sensitive area that requires expert supervision. Accordingly, machine learning methods should be interpretable or, at a minimum, provide explanations for their predictions. ADMIRE automatically extracts underlying temporal motifs that result in a future brain activity. Here, we take advantage of extracted motifs and use a decision tree to explain why a link is labeled as anomalous.

For each link $e = (u_1, u_2, r)$ in the network, we sample inter-view and intra-view walks starting from $w_0 \in \{u_1, u_2\}$, and then encode each walk (motifs) as discussed in Section 3.3. In the neural encoding phase motifs with similar temporal and structural patterns are expected to be close in the embedding space. Accordingly, given $k \geq 2$, we use a $k$-mean clustering algorithm (Lloyd, 1982) to cluster walks in the embedding space. Next, we construct feature vector $\mathbf{v}_e = \begin{pmatrix} p_1^1 & p_2^1 & \cdots & p_k^1 & p_1^2 & p_2^2 & \cdots & p_k^2 \end{pmatrix}$, where $p_i^1$ and $p_i^2$ are normalized counting number of sampled walks (motifs) starting from $u_1$ and $u_2$ in cluster $i$. That is, $p_i^1 = \frac{C_i^1}{C}$, where $C_i^1$ is the number of sampled walks starting from $u_1$ that are in cluster $i$, and $C$ is the total number of sampled walks. In this design, $p_i^1$s and $p_i^2$s describe the distribution of motifs in the neighborhoods of nodes $u_1$ and $u_2$, respectively. Now for each link in the data, we have a feature vector and a binary label assigned by Equation 4.

Sparse decision trees are one of the most popular forms of interpretable models (Rudin et al., 2022), and have shown competitive or better performance than blackbox models (McTavish et al., 2022). Since the importance of each motif (whether from an inter-view or intra-view walk) is different, we use a weighted optimal sparse decision tree model (Behrouz et al., 2022b), and train it on feature vectors $\mathbf{v}_e$s with binary labels assigned by Equation 4. This fast algorithm finds a weighted sparse decision tree provably close to the optimal tree, with arbitrary given depth, which provides both a performance guarantee and flexibility. For a given link $e$, this post-hoc explanation method can explain which motifs have important roles in ADMIRE's prediction on $e$. This process can help to expose the causes of abnormal brain activity in people living with a disease or disorder. For more details about optimal sparse decision tree and ADMIRE++ see Appendix D.

# 4. Experiments

**Datasets.** We use three real-world datasets: ① PD (Day et al., 2019) consists of the structural and the functional MRI images of 25 participants with and 21 participants without PD, who do the ANT task (Fan et al., 2005). The first view represents the fMRI, while the second view represents the T1-weighted structural MRI. ② ADHD (Brown et al., 2012) contains data for 50 subjects in the ADHD group and 50 subjects in the typically developed (TD) control group. Here, each view represents the brain network of an individual. ③ ASD (Craddock et al., 2013) contains data for 45 subjects in the ASD group and 45 subjects in the TD control group. The $i$-th view represents the brain network obtained by filtering the fMRI values in the range $[0.01 + (i-1) \times 0.02, 0.01 + i \times 0.02]$ Hz.

For the first part of the experiment, we follow the methodology used in existing studies (Akoglu et al., 2015; Behrouz & Seltzer, 2022) and synthetically inject anomalous edges into the brain networks in the control group (healthy or TD).

**Baselines.** We use ANOMULY (Behrouz & Seltzer, 2022), GOutlier (Aggarwal et al., 2011), NetWalk (Yu et al., 2018b), AddGraph (Zheng et al., 2019), ML-GCN (Behrouz & Hashemi, 2022), and MNE (Zhang et al., 2018b) as baselines in the transductive setting. Since there is no prior work on inductive learning in multiplex networks, we compare our model with inductive monoplex methods, CAW-N (Wang et al., 2021), TGAT (da Xu et al., 2020), and EvolveGCN (Pareja et al., 2020). For the detailed explanation of baselines see Appendix H.

## 4.1. Results on Synthetic Experiments

**Effectiveness Evaluation.** Table 1 reports the AUC for the edge anomaly detection task for the baselines and AD-MIRE. ADMIRE outperforms all baselines by a significant margin, $\min = 6.42\%$ and $\max = 18.04\%$ improvement in the transductive setting and with $\min = 11.08\%$ and $\max = 26.89\%$ improvement in the inductive setting. There are four reasons for ADMIRE's superior performance: ADMIRE is ① a multiplex method and can learn from different subjects, image modalities, or frequency bands. ② a stream-based method, using a time encoding module to capture *continuous* time information, while the baselines are snapshot-based and aggregate links, which removes useful time information (Wang et al., 2021). ③ scalable with respect to the number of views and can be trained on many data sources. ④ an end-to-end method with an exclusive design of architecture and generating negative samples for brain networks, while baselines are designed to learn the temporal and structural properties of a general network.

**Ablation Studies.** We next conduct an ablation study to val-

*Table 1.* Performance comparison (AUC).

| Methods | | PD | | ADHD | | ASD | |
|---|---|---|---|---|---|---|---|
| | Anomaly % | 1% | 5 % | 1% | 5 % | 1% | 5 % |
| | | | Monoplex Methods | | | | |
| Transductive | GOUTLIER | 61.42 | 59.98 | 65.37 | 64.70 | 60.85 | 59.13 |
| | NETWALK | 69.71 | 0.6902 | 70.29 | 69.86 | 69.07 | 68.52 |
| | ADDGRAPH | 71.94 | 70.33 | 71.89 | 70.11 | 71.30 | 70.96 |
| | | | Multiplex Methods | | | | |
| | MNE | 70.39 | 70.54 | 73.78 | 72.31 | 70.19 | 69.94 |
| | ML-GCN | 68.50 | 68.33 | -* | -* | 69.56 | 69.35 |
| | ANOMULY | 78.07 | 79.85 | -* | -* | 77.14 | 77.08 |
| | ADMIRE | **85.09** | **84.98** | **88.67** | **88.53** | **91.06** | **89.95** |
| Inductive | EvolveGCN | 55.18 | 55.06 | 57.23 | 57.41 | 56.89 | 56.21 |
| | TGAT | 59.34 | 58.72 | 60.19 | 60.10 | 60.28 | 59.93 |
| | CAW-N | 75.85 | 75.90 | 71.64 | 71.02 | 71.31 | 71.96 |
| | ADMIRE | **84.72** | **84.31** | **88.03** | **88.97** | **90.49** | **90.28** |

* Training time exceeds the threshold.

idate the effectiveness of each ADMIRE component. The results are summarized in Table 2. Rows 2 and 3 show the effectiveness of inter-view and intra-view walks. The ADHD dataset does not benefit from inter-view walks, because there is no causal relation between brain networks from different individuals, so inter-view walks are not informative, and our model should learn to ignore them (sets $\lambda = 0$). Rows 4 and 5 show the importance of the learnable parameter $\lambda$ and attention mechanism to incorporate information from different views. Rows 7, 8, and 9 show the importance of our new negative sample generator. When using an RNN instead of MLP-Mixer in the walk encoding phase (row 10), we gain better performance due to its ability to learn continuous time dynamics and the dependency of nodes' encodings in a walk. Finally, the last row shows the superior performance of multiplex ADMIRE over monoplex ADMIRE, when using only one brain network generated from a subject, image modality, or frequency band.

**Parameter Sensitivity.** We systematically analyze the effect of hyperparameters used in ADMIRE on the performance. Figure 2($a$) shows that only a small number of intra-view walks are enough to achieve competitive performance. A similar pattern can be seen for increasing the number of intra-view walks (Figure 2($b$)). Figure 2($c$) shows that ADMIRE might achieve the best performance at a certain walk length, while the exact value depends on the complexity of motifs that are required to learn underlying network dynamic law as well as the number of views. Finally, Figure 2($d$) shows the evolution of $\lambda$ in training. As expected, in datasets with no causal relationship between different views (e.g., ADHD), ADMIRE learns to set $\lambda \leq 0.1$ in a few numbers of epochs. For other datasets, it shows that ADMIRE converges very quickly to the best value of $\lambda$.

**Noisy Brain Images.** As we discussed in § 3.1, one of the main motivations for modeling neuroimaging datasets

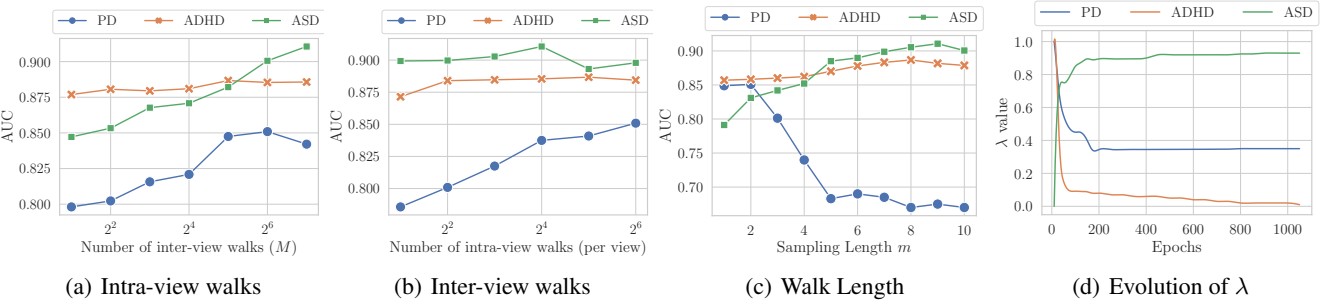

Figure 2. The effect of hyperparameters on the performance (a-c), and $\lambda$ evolution (d).

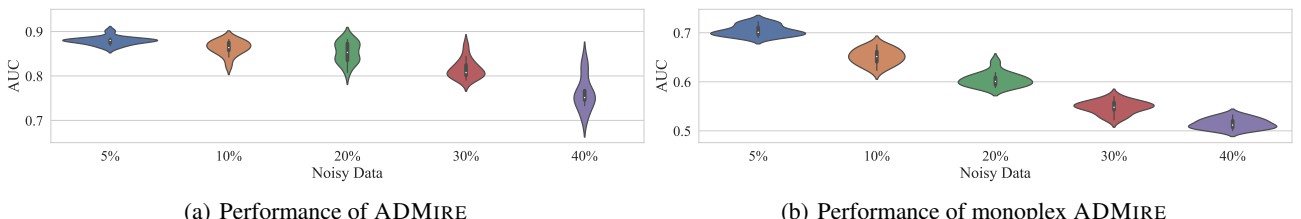

Figure 3. The advantage of multiplex brain networks over monoplex brain networks.

Table 2. Ablation study (AUC).

| | Methods | PD | ADHD | ASD |
|---|---|---|---|---|
| 1 | ADMIRE | **85.09** | **88.67**[*] | **91.06** |
| 2 | w/o inter-view | 78.59 | 88.73[*] | 80.65 |
| 3 | w/o intra-view | 77.14 | 69.59 | 79.62 |
| 4 | w/o $\lambda$ ($\lambda = 1$) | 80.42 | 80.36 | 89.30 |
| 5 | w/o attention | 84.79 | 86.14 | 86.57 |
| 6 | w/o time encoding | 84.16 | 82.78 | 85.92 |
| 7 | w/o inter-view NS | 84.77 | 84.28 | 83.46 |
| 8 | w/o intra-view NS | 79.91 | 78.75 | 81.09 |
| 9 | w/o historical NS | 84.68 | 84.16 | 84.31 |
| 10 | w/ RNN | 83.90 | 85.32 | 89.13 |
| 11 | Monoplex-ADMIRE | 76.52 | 72.07 | 74.15 |

[*] There is no causal relation between views.

as multiplex networks is to make the model more robust against noise in each brain image. To validate it, in this experiment, we add Gaussian noise to a subset of brain images ($5\%, 10\%, 20\%, 30\%$ and $40\%$) in the ADHD dataset. We model the noisy dataset as a multiplex brain network and use it to train ADMIRE. Next, as a baseline, following previous methods (Lanciano et al., 2020; Zhang et al., 2020), we take the average of all brain images in the noisy dataset and use it to train the monoplex ADMIRE. Figure 3 reports the performance of ADMIRE and monoplex AD-MIRE with varying the size of noisy samples. Not only ADMIRE achieves superior performance with a significant margin, but it also shows to be more robust against noise than the monoplex ADMIRE. This experiment shows the

importance of multiplex modeling and also the effectiveness of the proposed attention mechanism that can learn to ignore noisy samples.

### 4.2. Results on Real-world Datasets

We next train our model on the healthy control group and then test on the condition group to find anomalous brain activities of people in the condition group. Additional visualizations and results are in Appendix I.

**Parkinson's Disease.** We study how anomalous connections found by ADMIRE are distributed in the brain of people living with PD. Figure 4(a) reports the average distribution of anomalous edges in the brain networks of people living with PD. Most anomalous edges found by AD-MIRE have a vertex in either *Posterior Cingulate*, *Superior Parietal*, *Medial Orbitofrontal*, *Pars Opercularis*, or *Supramarginal Gyrus* ($\geq 95\%$ of all found anomalies). Next, we apply ADMIRE on the healthy control group to see whether these findings are exclusive to the PD group and to identify possible noise in the dataset. We observe that ADMIRE finds $94.2\%$ fewer anomalous connections in the healthy control group; most of these edges have a node in either *Temporal Pole* or *Anterior Insula*.

**Attention Deficit Hyperactivity Disorder.** Figure 4(b) shows the average distribution of anomalous edges in the brain networks of subjects in the condition ADHD group. Most abnormal connections found by ADMIRE have an

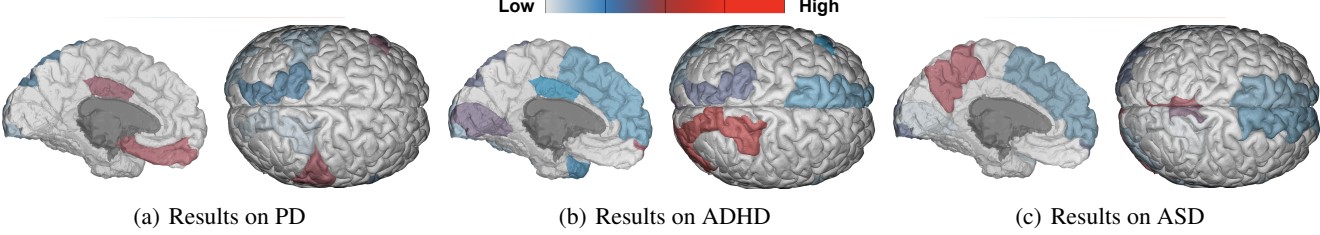

(a) Results on PD       (b) Results on ADHD       (c) Results on ASD

*Figure 4.* The distribution of anomalous edges in condition groups.

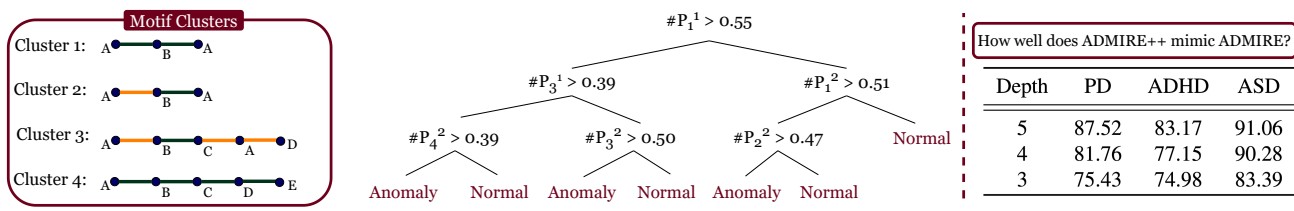

| How well does ADMIRE++ mimic ADMIRE? | | | |
|---|---|---|---|
| Depth | PD | ADHD | ASD |
| 5 | 87.52 | 83.17 | 91.06 |
| 4 | 81.76 | 77.15 | 90.28 |
| 3 | 75.43 | 74.98 | 83.39 |

*Figure 5.* (**Left**) Motif clusters and example of an extracted motif in each cluster. (**Middle**) The ADMIRE++ tree explanation on PD dataset (depth=3). (**Right**) Accuracy (%) of generated tree explanation.

endpoint in either *Frontal Pole*, *Right Lateral Occipital Cortex*, *Lingual Gyrus*, *Left Temporal Pole*, or *Right Superior Parietal Lobule* ($\geq 95\%$ of all found anomalies). Applying ADMIRE on the healthy control group, we observe that ADMIRE finds 89.6% fewer anomalous connections in the healthy control group; most of these edges have an endpoint in either *Planum Polare* or *Angular Gyrus*. Interestingly, these findings are consistent with previous studies on ADHD, using voxel-wise estimation of regional tissue volume changes (Wang et al., 2007), abnormality in DTI images (Lei et al., 2014), and Forman–Ricci curvature changes (Chatterjee et al., 2021), which shows the potential of ADMIRE in revealing abnormal connections that might be correlated to a brain disease or disorder.

**Autism Spectrum Disorder.** Figure 4(*b*) shows the average distribution in the brain networks in the ASD group. Most abnormal connections found by ADMIRE have an endpoint in either *Right Superior Temporal Gyrus*, *Right Cerebellum Cortex*, *Right Precuneus*, *Frontal Pole*, *Left Lateral Occipital* ($\geq 95\%$ of all found anomalies). Applying ADMIRE on the healthy control group, ADMIRE finds 93.7% fewer anomalous connections in the healthy control group, most of which have an endpoint in either *Temporal Pole* or *Posterior Cingulate Cortex*. Although several works have studied ASD and found different abnormality patterns, there is still no known ASD biomarker (Müller & Linke, 2021). However, a part of our findings about the abnormal activity in the cerebellum cortex is consistent with previous studies (Rogers et al., 2013).

**ADMIRE++ Explanations.** To evaluate the quality of AD-

MIRE++, we use the PD dataset and set $k = 4$, so we have 4 clusters and 8 features for each link (4 features for each of its endpoints). Figure 5 (left) and (middle) show an example of motifs in each cluster and a decision tree with depth 3 that mimics the ADMIRE's predictions. While motifs in cluster 1 and 2 shows that the neighborhood of a node is sparse, motifs in cluster 3 and 4 show that the neighborhood of the node is dense. One can interpret the decision tree prediction as: a link is normal if the neighborhoods of its endpoints are both sparse or dense and is abnormal otherwise. The table on the right reports the accuracy of how well ADMIRE++ mimics ADMIRE's predictions. Even with small depths, ADMIRE++ produces explanations with high accuracy.

## 5. Conclusion

We present ADMIRE, an end-to-end inductive unsupervised learning method on multiplex networks to detect abnormal brain activity that might suggest a brain disease or disorder. ADMIRE uses inter-view (resp. intra-view) temporal walks to implicitly extract network motifs and causal relationships across different views (resp. within a view) and adopts novel anonymization based on the correlation between network motifs to hide the identity of nodes and views. Next, it uses an MLP-Mixer to encode the sequence of nodes in a walk. To explain its prediction, we design ADMIRE++, a post-hoc decision-tree-based method that explains ADMIRE's predictions via extracted motifs. Our experimental results show the superior performance of ADMIRE against baselines and the potential of ADMIRE in detecting abnormal brain activity, undetected in previous studies.

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

## A. Reproducibility

The implementation of ADMIRE is available in `https://github.com/ubc-systopia/ADMIRE`.

## B. Additional Related Work

To situate our research in a broader context, we briefly review research in ① temporal graph learning methods, ② multiplex graph learning, ③ feature learning in brain networks, ④ anomaly detection in brain networks, and ⑤ anomaly detection in multiplex networks.

**Temporal Graph Learning.** Learning from temporal networks has been a widely studied topic in the literature (Longa et al., 2023). The first group of methods uses a Graph Neural Network (GNN) as a feature encoder and then uses a sequence model on top of the GNN to capture temporal properties (Peng et al., 2020; Wang et al., 2020b; Yu et al., 2018a). The second group uses Recurrent Neural Networks (RNNs) with a GNN layer replacing the linear layer to learn from the temporal network (Li et al., 2018; Seo et al., 2018; Zhao et al., 2019; You et al., 2022; Hashemi et al., 2023). Recently, more conceptually complicated learning methods for temporal graphs have been designed based on temporal random walks (Wang et al., 2021; Jin et al., 2022; Behrouz et al., 2023), line graphs (Chanpuriya et al., 2023), neighborhood representation (Luo & Li, 2022), and subgraph sketching (Chamberlain et al., 2023). Cong et al. (2023) design a simple but effective temporal edge encoding method and show that self-attention mechanisms and RNNs are not essential for temporal graph learning. However, all these methods differ from our approach as they are designed for simple temporal graphs and cannot easily be extended to graphs with different types of edges (multiplex networks).

**Multiplex Graph Learning.** In the literature, multiplex networks (also known as multi-view, multilayer, or multi-dimensional networks) are graphs with a node type but multiple edge types (relations) (Kivelä et al., 2014). Several methods have been proposed to learn network embeddings on multiplex networks by integrating information from individual relation types (Cen et al., 2019; Pio-Lopez et al., 2021; Yan et al., 2021; Chang et al., 2015; Xie et al., 2021; Wang et al., 2020a). Other work proposed Graph Convolutional Networks (GCNs) methods for multiplex networks (Behrouz & Hashemi, 2022; Cheng et al., 2021; Zhang et al., 2018a). Inspired by Deep Graph Infomax (Veličković et al., 2019), Park et al. (2020) and Jing et al. (2021) proposed unsupervised approaches to learn node embeddings by maximizing the mutual information between local patches and the global representation of the entire graph. Zhang et al. (2018b) proposed a method that uses a latent space to integrate the information across multiple views. Recently, Wang et al. (2022) proposed DPMNE to learn from incomplete multiplex networks. All these methods are designed in the transductive setting for static multiplex networks, which is different from our formulation.

**Feature Learning in Brain Networks.** In recent years, several studies focused on analyzing brain networks to understand and distinguish healthy and diseased human brains (Jie et al., 2016; Chen et al., 2011; Wee et al., 2011). Recently, due to the success of GNNs in analyzing graph-structured data, deep models have been proposed to predict brain diseases by learning the graph structures of brain networks (Kan et al., 2021; Cui et al., 2021; Kan et al., 2022a; Zhu et al., 2022a; Cui et al., 2022b). All these methods are designed for the graph or node classification and cannot easily be extended to the edge-anomaly detection task.

**Anomaly Detection in Brain Networks.** In addition to predicting disease in brain networks, understanding the cause of the disease is important. To this end, several anomaly detection methods have been proposed to find anomalous connections, regions, or subgraphs in the brain, which can cause a disease (Chatterjee et al., 2021; Zhang et al., 2016; Liu et al., 2020). All these methods are designed for node or subgraph anomaly detection tasks and cannot easily be extended to the edge-anomaly detection task.

**Anomaly Detection in Multiplex Networks.** The problem of anomaly detection in multiplex networks has recently attracted attention. Mittal & Bhatia (2018) use eigenvector centrality, page rank centrality, and degree centrality as handcrafted features for nodes to detect anomalies in static multiplex networks. Bindu et al. (2017) proposed a node anomaly detection algorithm in static multiplex networks that uses handcrafted features based on clique/near-clique and star/near-star structures. Bansal & Sharma (2020) defined a quality measure, Multi-Normality, which uses the structure and attributes together of each view to detect attribute coherence in neighborhoods between layers. Maulana & Atzmueller (2020) use centrality of all nodes in each view and apply many-objective optimization with full enumeration based on minimization to obtain Pareto Front. Then, they use Pareto Front as a basis for finding suspected anomaly nodes. Chen et al. (2022) proposed

---

**Algorithm 1** Temporal multiplex walk sampling procedure

---

**Require:** The edge set $\mathcal{E}$, previously sampled node $w_p$ in view $r_p$ at time $t$, and hyperparameter $\mu$
**Ensure:** Next sampled connection $(w_n, w_p, r_n, t_n)$
1: **for** $e = (w_n, w_p, r_n, t_n) \in \mathcal{E}^{t_p}(w_p)$ **do**
2:      Sample $b \sim$ UNIFORM$(0, 1)$;
3:      **if** $b < q_{r_p}^{t_p}(w_p, e) \times \varphi(r_p, r_n)$ **then**
4:          **return** $e = (w_n, w_p, r_n, t_n)$;
5:      **end if**
6: **end for**
7: **return** EOA;

---

ANOMMAN that uses an auto-encoder module and a GCN-based decoder to detect node anomalies in static multiplex networks. Although this model can learn from the data, it is limited to static networks, and it treats each view equally in the *Structure Reconstruction* step. Finally, Ofori-Boateng et al. (2021) developed a new persistence summary and used it to detect events in dynamic multiplex blockchain networks. All of these approaches are designed to detect topological anomalous subgraphs, nodes, or events, and cannot identify anomalous edges. Moreover, these methods, except ANOMAN (Chen et al., 2022), are based on pre-defined patterns/roles or handcrafted features, while real-world network anomalies have complex nature. Therefore, these models cannot be generalized to different domains, limiting their application.

The only exception and also the closest method to our approach is ANOMULY (Behrouz & Seltzer, 2022), a GNN-based anomaly detection method in multiplex networks. However, this method suffers from four main limitations: ① Transductive learning: The ANOMULY framework is designed in a transductive setting and cannot be applied to unseen nodes/patterns. In contrast, ADMIRE anonymizes nodes in such a way to work in the inductive setting. ② Memory and scalability: The ANOMULY framework is snapshot-based. That is, it requires storing the entire snapshot of the temporal network at each timestamp, which consumes a great deal of memory. Moreover, since it uses different GNN modules for each type of connection, it cannot be utilized for multiplex brain networks with a large number of views (e.g., in datasets with a large number of participants). However, ADMIRE is a streaming method, requiring only constant memory (see Appendix C). Moreover, our random walk encoder scales to brain multiplex networks with more than 100 views. ③ Lack of generalizability: The ANOMULY framework uses a simple negative sampling method by randomly changing one endpoint of a connection to learn anomalous interactions. While this negative sampling method is fast and lets the model be trained in an unsupervised manner, these negative sample generator methods are too simple and can cause poor performance in more complicated datasets (Poursafaei et al., 2022). ADMIRE introduces a novel negative sampling method for multiplex networks and shows its efficacy in § 4.1. ④ Many hyperparameters: The ANOMULY framework has many hyperparameters that require tuning before the model achieves good performance. However, tuning these hyperparameters is difficult in real-world datasets, limiting its applications. In ADMIRE, there are only four hyperparameters, which can simply be tuned based on the dataset properties.

## C. Efficient Sampling

The first step in our sampling is to compute the sampling probability of an incoming connection in relation type $r$. For an incoming edge $e = (u, v, r, t)$ we compute the probabilities $q_r^t(w)$ for $w \in \{u, v\}$ as follows:

$$q_r^t(w, e) = \frac{\exp\left(\mu t\right)}{\sum_{(w_0, t') \in \mathcal{N}_r^t(w)} \exp\left(\mu t'\right)}, \tag{5}$$

where $\mathcal{N}_r^t(w)$ represents the set of $w$'s neighbor in view $r$ and before time $t$. This probability needs to be computed one time when arrives and does not need to be updated anymore. Also, for calculating the probability of sampling this connection after a connection from another relation type $r'$, we simply multiply this probability by $\varphi(r, r')$.

Algorithm 1 shows the sampling procedure. Given a previously sampled connection in view $r$ at time $t$, we sample the next connection in view $r'$ at tme $t' < t$ with a probability proportional to $\exp\left(\mu(t' - t)\right) \times \varphi(r, r')$. It is not hard to show that Algorithm 1 sample the next connection with a probability proportional to $\exp\left(\mu(t_n - t_p)\right) \times \varphi(r_p, r_n)$. Inspired by Wang et al. (2021), in our experiments, we store most $k$ recent connections with $k \propto \mathcal{O}\left(\frac{1}{\mu}\right)$. The intuition is that if we sort connections in $\mathcal{E}^t(w_p)$ by their timestamp $\{t_i\}_{i=1}^h$, and assume that $\exp\left(\mu(t_i - t)\right)$ are i.i.d., the probability of sampling

$j$-th connection is:

$$\mathbb{P}[\text{sampling } j\text{-th connection}] = \frac{\prod_{i=1}^{j} \exp\left(\mu(t_i - t)\right) \times \varphi(r_p, r_i)}{\sum_{i=1}^{h} \prod_{s=1}^{i} \exp\left(\mu(t_s - t)\right) \times \varphi(r_p, r_s)}.$$

It is not hard to see that this probability is very small when we increase the value of $j$. Accordingly, in practice, we only need to store a constant number of the most recent connections at each time.

## D. Optimal Sparse Decision Trees and ADMIRE++

**Optimal Sparse Decision Trees.** Sparse decision trees, a prominent category of interpretable machine learning models, are widely utilized for decision making (e.g., Ernst et al., 2005; Silva et al., 2020). Traditionally, the optimization of decision trees involved a greedy approach, constructing trees from the top down (Quinlan, 1993; Breiman et al., 1984). However, recent advancements have introduced several methods that thoroughly optimize sparse trees to achieve an optimal balance between performance and interpretability (Farhangfar et al., 2008; Bertsimas & Dunn, 2017; Günlük et al., 2021). It is worth noting that optimizing sparse optimal trees is a computationally challenging task, with NP-hard complexity. Nevertheless, recent research has exploited the discrete nature of the loss function, resulting in computational advantages (Aghaei et al., 2021; McTavish et al., 2022; Lin et al., 2020). Recently, to optimize decision trees over weighted datasets, where each sample has a weight, Behrouz et al. (2022b) design three efficient algorithms to optimize weighted sparse decision trees. Let $\mathcal{T}$ be a decision tree that gives predictions $\{\hat{y}_i^{\mathcal{T}}\}_{i=1}^{N}$ on dataset $(\mathbf{x}, \mathbf{y})$. The weighted loss of $\mathcal{T}$ is defined as:

$$\mathcal{L}_{\mathbf{w}}(\mathcal{T}, \mathbf{x}, \mathbf{y}) = \frac{1}{\sum_{i=1}^{N} w_i} \sum_{i=1}^{N} \mathbb{1}[y_i \neq \hat{y}_i^{\mathcal{T}}] \times w_i. \tag{6}$$

The algorithm aims to maximize the loss with depth and sparsity constraints:

$$\underset{\mathcal{T}}{\text{minimize}} \ \mathcal{L}_{\mathbf{w}}(\mathcal{T}, \mathbf{x}, \mathbf{y}) + \lambda H_{\mathcal{T}} \quad s.t. \ \text{depth}(\mathcal{T}) \leq d, \tag{7}$$

where $H_{\mathcal{T}}$ is the number of leaves in $\mathcal{T}$ and $\lambda$ is a per-leaf regularization parameter. It is theoretically proven that their algorithms can find a decision tree very close to the globally optimal decision tree, which guarantees the performance of found tree.

**ADMIRE++.** We design a post-hoc explainability method to mimic the predictions of ADMIRE. The main intuition of ADMIRE is to extracts causality of network dynamic by backtracking over time and extracts network multiplex motifs. We use these extracted motifs to describe the neighborhood of each node and then use this data to train a decision tree model. Accordingly, the decision tree can explain why a connection is abnormal based on the neighborhoods of its endpoints. However, the main challenge is that we have diverse network motifs in large and dense networks like brain networks, and accordingly, we need so many features to accurately describe the neighborhood of each node. To this end, we apply $k$-mean clustering algorithm (Lloyd, 1982) to the walk embeddings obtained in subsection 3.3, and group network motifs based on their similarity. Clearly, the larger $k$ results in a better explanation. Next, to construct features that we can use to train the decision tree, we define $\mathbf{v}_e = \begin{pmatrix} p_1^1 & p_2^1 & \dots & p_k^1 & p_1^2 & p_2^2 & \dots & p_k^2 \end{pmatrix}$, where $p_i^1$ and $p_i^2$ are normalized counting number of sampled walks (motifs) starting from $u_1$ and $u_2$ in cluster $i$. That is, $p_i^1 = \frac{C_i^1}{C}$, where $C_i^1$ is the number of sampled walks starting from $u_1$ that are in cluster $i$, and $C$ is the total number of sampled walks. Similarly, $p_i^2 = \frac{C_i^2}{C}$, where $C_i^2$ is the number of sampled walks starting from $u_2$ that are in cluster $i$. Therefore, in this design, $p_i^1$s and $p_i^2$s describe the distribution of motifs in the neighborhoods of nodes $u_1$ and $u_2$, respectively. For each link $e$ in the data, we have a feature vector $\mathbf{v}_e$ and a binary label assigned by Equation 4. Accordingly, we train our decision tree on these features and labels via algorithm proposed by Behrouz et al. (2022b). The sparsity in optimal sparse decision tress guarantees the interpratability and also the algorithm proposed by Behrouz et al. (2022b) guarantees the performance quality of the trained model.

Note that, to assign a weight to each sample, which here is a link, we use the inner product of its type encoding and the average of its endpoints. That is, given $e = (u_1, u_2, r)$:

$$\mathbf{w}_e = \mu(r).\left(\frac{\zeta(u_1) + \zeta(u_2)}{2}\right), \tag{8}$$

where $\mu(r)$, $\zeta(u_1)$, and $\zeta(u_2)$ are calculated in subsection 3.3.

# E. Attention Mechanism: Motivation

As we discussed, in multiplex networks the importance of views might be different. For example, one disease might be more correlated with functional connectivity than structural connectivity, or a brain network of an individual can be noisy and we need to automatically ignore it in the training process. To this end, we design an attention mechanism that learns the importance of each view for other views. Existing attention mechanisms (Behrouz & Seltzer, 2022; Park et al., 2020) are designed for *general* multiplex networks, assuming that the importance of each view for each node is different. Although these mechanisms are more general, they are required to learn many parameters, limiting their scalability to large networks with a large number of views. Here, in our experimental evaluations on multiplex *brain* networks, we observe that the importance of one view for different nodes is almost the same. Given a view $r$ and a node $u$, we use $\Omega(u, r)$ to show the importance of view $r$ for node $u$. We use the attention mechanism proposed by Behrouz & Seltzer (2022) instead of our attention mechanism and train the model on PD, ADHD, and ASD datasets. While it requires $\approx 1.8\times$ training time, we observe that $\Omega(u, r) \approx \Omega(v, r)$ for any given view $r$ and arbitrary nodes $u$ and $v$. That is, given a view $r$, the maximum variance of $\Omega(u, r)$ for different nodes $u$ is 0.02, 0.05, and 0.02 in PD, ADHD, and ASD datasets, respectively. Therefore, since in a multiplex brain network we might have a large number of views (e.g., a large number of subjects, a large number of image modalities, or a large number of frequency bands), we design a more efficient and scalable attention mechanism that learns the importance of each view for other views (independent of nodes). One can interpret this attention mechanism as a model that learns the correlation between each pair of views.

# F. Experimental Setting Details

We tune hyper-parameters by cross-validation, and search the hyper-parameters over ① $\mu \in \{0.5, 1, 2, 4\} \times 10^{-5}$, ② Inter-view sampling number $M \in \{32, 64, 128, 256\}$, ③ Intra-view sampling number per view $M' \in \{8, 16, 32, 64\}$, ④ Walk length $m \in \{2, 4, 8, 12\}$. Also, in the training, we use a learning rate of 0.0001, hidden dimension 100 in MLP-Mixer, and batch size of 600.

To visualize the average distribution of anomalous connections, we use BrainPainter (Marinescu et al., 2019) with the Desikan-Killiany atlas.

# G. Datasets

We evaluate ADMIREusing three real-world datasets, PD (Day et al., 2019), ADHD (Brown et al., 2012), and ASD (Craddock et al., 2013), as well as three synthetic datasets. Each of the datasets represents one type of multiplex brain network modeling proposed in §3.1.

**PD Dataset.** Attention dysfunction is a common symptom of Parkinson's disease (PD) and has a significant impact on quality of life. This dataset (Day et al., 2019) uses the Attention Network Test (ANT) (Fan et al., 2005) and is designed to study three aspects of attention: alerting (maintaining an alert state), executive control (resolving conflict), and orienting. It consists of structural and functional MRI images of participants with and without PD, with six repetitions of the ANT task (Fan et al., 2005). It contains data for 25 subjects (7 female, age = $66.1 \pm 10.0$ yrs, years since disease onset = $8.4 \pm 4.8$) in the PD group and 21 subjects (12 female, age = $62.1 \pm 9.9$ yrs) in the healthy control group. We model the data using a temporal multiplex brain network with two views, 114 ROIs, and six timestamps (fMRI during each task). The first view represents the brain network obtained from fMRI, while the second view represents that generated from T1-weighted structural MRI.

**ADHD Dataset.** This dataset consists of resting fMRI data taken from USC Multimodal Connectivity Database (USCD) (Brown et al., 2012). The dataset contains data for 50 subjects (27 female, age = $9.84 \pm 3.57$ yrs) in the ADHD group and 50 subjects (25 female, age = $12.74 \pm 4.1$ yrs) in the typically developed (TD) control group. This dataset is preprocessed [1]. We model this data using a temporal multiplex brain network with 50 views, 190 ROIs, and 10 timestamps; each view represents the brain network of an individual.

**ASD Dataset.** This dataset consists of resting fMRI data taken from the Autism Brain Imaging Data Exchange (ABIDE) (Craddock et al., 2013); it contains data for 45 subjects (23 female, age = $23.1 \pm 8.1$ yrs) in the ASD group

---

[1] https://ccraddock.github.io/cluster_roi/atlases.html

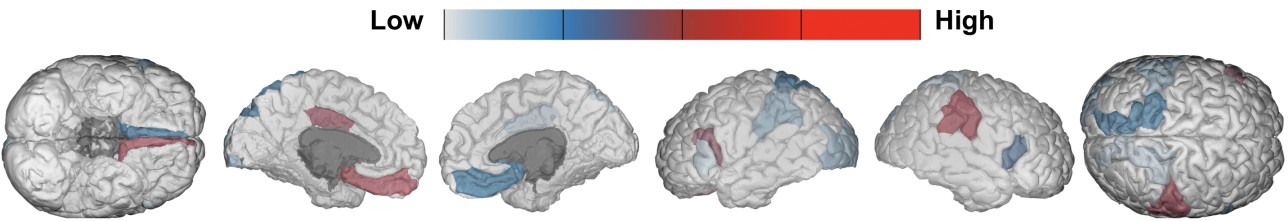

*Figure 6.* The distribution of anomalous edges in PD group.

and 45 subjects (22 female, age $= 25.4 \pm 8.9$ yrs) in the typically developed control group. We have followed the five pre-processing strategies denoted as DPARSF, followed by Band-Pass Filtering with different filters in a range between 0.01 and 0.25 Hz, in steps of 0.02 Hz. This range and steps are previously motivated by (De Domenico et al., 2016). We model this data using a temporal multiplex brain network with 12 views, 116 ROIs, and 10 timestamps; the $i$-th view represents the brain network obtained by filtering the fMRI values in the range $[0.01 + (i - 1) \times 0.02, 0.01 + i \times 0.02]$ Hz.

**Synthetic Datasets.** We use synthetic datasets to show ① the effectiveness of ADMIRE in detecting anomalous connections compare to baselines, ② the importance of each element in our framework (ablation study), and ③ the advantage of modeling brain images as multiplex networks compared to modeling them as monoplex networks. Since the ground truth label for anomaly detection (specifically in brain networks) is difficult to obtain (Akoglu et al., 2015) (ground truth is unknown in many real neuroimaging data), we follow the methodology used in existing studies (Akoglu et al., 2015; Zheng et al., 2019; Yu et al., 2018b; Behrouz & Seltzer, 2022) and synthetically inject anomalous edges into our brain networks of people in the control group (healthy or TD) from our datasets. Accordingly, the nature of our synthetic datasets is real brain networks; however, synthetically anomalous connections are added to mitigate the lack of labeled data.

**Pre-pocessing.** Unless stated otherwise, for preprocessing and constructing brain networks from original fMRI and DTI data, we use the FSL toolbox and BrainGB (Cui et al., 2022a). Each edge in the fMRI brain networks shows that the statistical correlation between its endpoint is more than $80$-th percentile of the distribution of correlation values.

## H. Baselines

Since ANOMULY (Behrouz & Seltzer, 2022) is the only competitor method on edge anomaly detection in multiplex networks, we also compare ADMIRE with single-layer edge anomaly detection methods: GOutlier (Aggarwal et al., 2011) builds a generative model for edges in a node cluster. NetWalk (Yu et al., 2018b) uses a random walk to learn a unified embedding for each node and then dynamically clusters the nodes' embeddings. AddGraph (Zheng et al., 2019) is an end-to-end approach that uses an extended GCN in temporal networks. Finally, we compare with two multiplex network embedding baselines, ML-GCN (Behrouz & Hashemi, 2022) and MNE (Zhang et al., 2018b). We apply $K$-means clustering on their obtained node embeddings for anomaly detection (Yu et al., 2018b).

In the inductive setting, since there is no inductive learning (or anomaly detection) method on multiplex networks that we are aware of, we compare ADMIRE with inductive learning methods on monoplex networks. CAW-N (Wang et al., 2021) is an inductive method that uses causal anonymous walks to extract network motifs and a novel set-based anonymization process that keep model inductive by hiding the identity of nodes during the training phase. EvolveGCN (Pareja et al., 2020) uses a RNN to estimate the GCN parameters for the future snapshots. TGAT (da Xu et al., 2020) uses GAT (Veličković et al., 2018) to extract node representations where the nodes' neighbors are sampled from the history and then encodes temporal information via random Fourier features.

## I. Additional Results on Real-world Datasets

In this section, we present additional visualization of results provided in § 4.1. Figure 6, Figure 7, and Figure 8 present the average distribution of anomalous edges in PD, ADHD, and ASD groups.

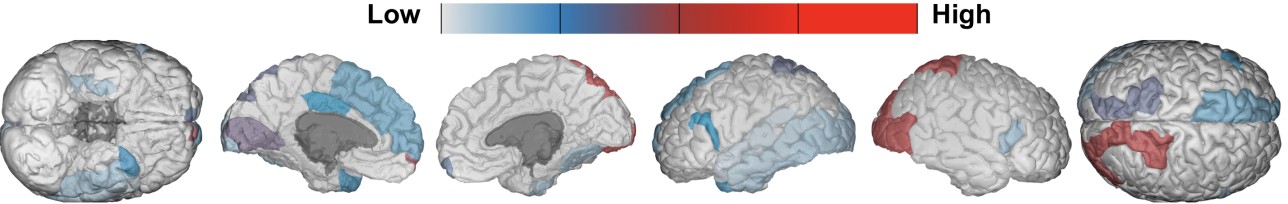

Figure 7. The distribution of anomalous edges in ADHD group.

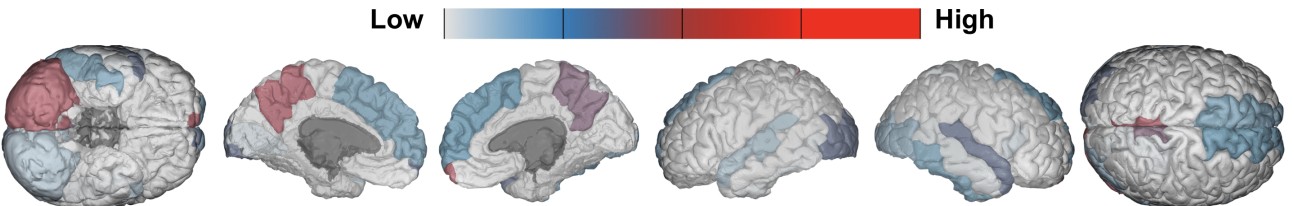

Figure 8. The distribution of anomalous edges in ASD group.

