# OpenReview forum: "ADMIRE++: Explainable Anomaly Detection in the Human Brain via Inductive Learning on Temporal Multiplex Networks"
_ICML.cc/2023/Workshop/IMLH — IMLH 2023 Oral_

### Official Review · Reviewer_rppR · 2023-06-12
**An end-to-end framework to detect anomalous brain activity with interpretability**

**Rating:** 8
**Confidence:** 2

**Review:**

Summary: This long paper presents an end-to-end framework for anomaly detection of brain networks. This ADMIRE approach utilizes both inter-views and intra-views temporal walk to gather node information. It incorporates walks' encodings to compute the anomaly score. Based on the prediction, the paper proposed a post-hoc explanation for temporal motifs that result in anomalous brain activity.

Novelty: Although I am not familiar with the content, the tackled problem is important and interesting. The methodology sounds novel to me.

Soundness: The experiments are comprehensive and well-explained. It demonstrates the efficacy of the proposed approach for both anomaly detection and interpretation.
- Is there any intuition behind using random walk in this framework besides the empirical verification?
- It would be helpful to add more clinical analysis for real-world results (section 4.2).

Clarity/Quality of writing: The paper is well-written and easy to follow. The structure of the paper is clear.
- It would be helpful to include the graph size for the experiments.

Strength:
1. Integrate both inter-views and intra-views in the random walk for anomaly detection on brain networks.
2. Provide interpretability with this anomaly detection method.
2. Evaluate this approach on various real-world healthcare dataset and illustrates how the interpretability of the ADMIRE can help in a clinical setting.

---

### Official Review · Reviewer_yqNj · 2023-06-17
**The article provides a novel graph ML-based  framework for brain anomaly detection. It addresses multiple limitations of the existing methods and achieve SOTA performance.  More details and comments are included in the Strength and Weaknesses section.**

**Rating:** 9
**Confidence:** 4

**Review:**

**Summary Of The Paper:**

This paper introduces ADMIRE (Anomaly Detection in Multiplex Brain Networks) to detect abnormal brain activity that might be attributed to brain diseases. First, most existing graph ML-based methods are either monoplex (designed for one modality or one individual subject) or transductive (assuming predefined patterns). While modeling neuroimages as multiplex brain networks with different types of edges, ADMIRE uses two temporal walks, inter-view and intra-view, to capture casual relations between and within views over time. It also uses the anonymization process to hide the identity of nodes and edges to keep the model inductive during training. The framework also uses a new attention mechanism to exploit complementary information from different views, a non-periodic time encoding, and an MLP-Mixer to learn the structural and temporal properties of the network. Finally, the authors employ k-means clustering and a decision tree-based method (ADMIRE++) to explain ADMIRE’s predictions using extracted motifs. The proposed framework achieves the SOTA performance over other methods.

**Strength And Weaknesses:**

DISCLAIMER: I am not deeply familiar with the related literature.

Strengths:

1.	The paper is well-written and makes a strong case.
2.	Most of the key components, such as inter-walk, intra-walk, modified anonymization, and attention mechanism, seem novel and are explained well with intuition and derivations.
3.	SOTA performance over other baselines.
4.	The predictions are explained with a tree-based method to give more insights into how decisions are made.
5.	Experiments are conducted in both synthetic and real settings.

Weaknesses:

1.	In Section 3.3,  the training strategy – negative sample generation, could be explained in more detail and why using this method. Similarly, for margin-based pairwise loss, an equation would be preferred. Perhaps include these in the appendix.
2.	It would be better to extend the discussion for results on real-world datasets, especially for Parkinson’s Disease, a little more to increase the clinical relevance.

Comments

1.	In Table 1, why are there results for both transductive and inductive ADMIRE? Transductive ADMIRE seems to have better results, which are also presented in Table 2 for the ablation study.
2.	Given the accuracy of ADMIRE++ in Figure 4,  with depth = 5, it seems to outperform ADMIRE, which could be addressed in the text.

**Clarity, Quality, Novelty, And Reproducibility:**

The paper is written clearly and intuitively. Detailed comments related to quality and novelty are provided in the Strength section. The authors claim that they will provide the repository link and more explanation of implementation details for reproducibility.

---

### Meta-Review · Area_Chair_vksH · 2023-06-20

**Recommendation:** Accept (Oral)
**Confidence:** 5

**Metareview:**

Both reviewers hold positive views on this paper, acknowledging its clarity and extensive experiments. I kindly ask the authors to consider the shortcomings pointed out by the reviewers and address these issues in the final version.

---

### Decision · Program_Chairs · 2023-06-20

Accept (Oral)